

# Reviews and syntheses: Tufa microbialites on rocky coasts towards an integrated terminology

Thomas W. Garner[1], J. Andrew. G. Cooper[1,2], Alan M. Smith[2], Gavin M. Rishworth[3], Matt Forbes [4]

[1]School of Geography and Environmental Science, Ulster University, Coleraine, BT52 1SA, Northern Ireland
[2]Discipline of Geology, School of Agriculture, Earth and Environmental Sciences, University of KwaZulu-Natal, P. Bag X54001 Durban, 4000, South Africa
[3]Department of Zoology, Nelson Mandela University, Gqeberha (Port Elizabeth), 6031, South Africa
[4]ARC Centre for Excellence for Australian Biodiversity and Heritage, University of Wollongong, Australia

*Correspondence to*: Thomas W. Garner (Garner-T@ulster.ac.uk)

**Abstract.** Microbialites are known from a range of terrestrial, freshwater, marine, and marginal settings with the applied descriptive terminology depending largely on the historical legacy derived from previous studies in similar environmental settings. This has led to a diversity of nomenclature and a lack of conformity in the terms used to describe and categorise microbialites. As the role of microbial mats and biofilms is increasingly recognised in the formation of tufa and terrestrial carbonates, deposits such as tufa microbialites bridge the spectrum of microbialites and terrestrial carbonate deposits.

Groundwater spring-fed tufa microbialites in supratidal rock coast environments occur at the interface of terrestrial and marine domains and necessitate the adoption of an integrative and systematic nomenclature approach. To date, their global distribution and complex relationships with pre-defined deposits have resulted in the application of a variety of descriptive terminologies, most frequently at the macro- and meso-scale. Here we review and consolidate the multi-scale library of terminologies for microbialites and present a new geomorphological scheme for their description and classification. This scheme has greater alignment with terrestrial carbonate nomenclature at the macroscale and with marine and lacustrine microbialites at the
mesoscale. The proposed terminology can primarily be applied to tufa microbialites in spring-fed supratidal settings but may also be applied to other relevant environmental settings, terrestrial carbonates, microbial mats and other microbialites.

## 1 Introduction

Burne and Moore (1987) defined microbialites as "*organosedimentary deposits that have accreted as a result of a benthic
*microbial community trapping and binding detrital sediment and/or forming the locus of mineral precipitation*". They have been described from a diverse range of depositional environments in both fossil and modern settings. These include marine settings with hypersaline conditions, (e.g. Hamelin Pool, Western Australia (Jahnert and Collins, 2012, 2011; Logan, 1961; Logan et al., 1964; Suosaari et al., 2022, 2019a, 2016)), normal salinity conditions (e.g., Bahamas (Dill et al., 1986; Reid et al., 1995)), and a range of non-marine environments including, but not limited to: hyposaline (e.g., Moore and Burne, 1994);
freshwater fluvial (e.g., Arenas et al., 2019; Caudwell et al., 2001); open lacustrine (e.g., Doddy et al., 2019); ice-covered





lacustrine (e.g., Mackey et al., 2015); hot springs (e.g., Walter et al., 1976) and caves (e.g., Brunet and Revuelta, 2014; Lundberg and McFarlane, 2011). Marine and non-marine forms have traditionally been studied independently and a distinct suite of definitions and nomenclature has been applied in each setting. As noted by Shapiro (2005, p.73) "[because] *universal terminology have not been ratified by the general community, clear descriptions will facilitate future research*". Recently, the

Handbook for the study and description of microbialites (Grey and Awramik, 2020) attempted to consolidate and refine the terminology applied to microbialites. This, however, explicitly excluded tufa microbialites and microbial mats from consideration due to these lacking detailed morphological description. This includes rock coast microbialites, which along with other tufa microbialites and microbial mats are termed 'non-classical' microbialites here.

Microbialites on high-energy rock coastlines have affinities with both terrestrial and marine forms (Rishworth et al., 2020), and their study brings into sharp focus the need for unifying terminology. To date, the terminology applied to rock coast microbialites has been varied and often complex. This is due to the application of terminology from various research areas including 'classical' microbialites (both fossil and contemporary), 'non-classical' microbialites (including tufa microbialites and microbial mats) and terrestrial carbonates, predominantly tufa. Their apparently scattered distribution in widely separated

locations led to variability in the descriptive terminology applied by independent research groups.  Now that they are known to have a global distribution, there is a need for clear and unambiguous terminology.

At the nexus of terrestrial and marine domains, rock coast microbialites afford the opportunity to assess the utility of existing terminology from both fields of research, to identify *inter alia* duplication and redundancy, and in so doing to present a unifying

terminology. This study specifically aims to:
-   Review the classification and description of rock coast microbialites using different disciplinary approaches.
-   Review and integrate terrestrial and marine microbialite terminology applied to rock coast settings.
-   Produce a synthesised geomorphological terminology for rock coast microbialites.

## 2 Defining rock coast microbialites

The term 'rock coast' is applied here to describe a specific depositional environment of "*rock coast settings in high intertidal to supratidal locations that are affected by occasional marine storms and/or high swells and the effects of sea spray*" (Cooper et al., 2022). Generally, rock coasts are characterised by resistant bedrock and/or hardground superficial deposits. After a historic description at Bonza Bay, South Africa by Mountain (1937), the first modern rock coast microbialite was documented by Smith and Uken (2003) from supratidal pools at Kei River mouth (Cape Morgan) on the south-eastern coastline of South

Africa. Subsequent occurrences were reported along the southern African coastline from Port Elizabeth to Tofu, Inhambane, Mozambique (Perissinotto et al., 2014; Smith et al., 2011); Southern and Western Australia (Forbes et al., 2010; Lipar and Webb, 2015), Northern Ireland, Scotland and Ireland (Cooper et al., 2022, 2013; Smith et al., 2018).



Rock coast microbialites are considered a tufa microbialite, "*a tufa forming as a result of microbial activity*" (Grey and
Awramik, 2020) with tufa being defined as the "*products of calcium carbonate precipitation under cool, ambient temperature
freshwater*" (Pedley, 1990). Tufa is increasingly considered to have significant biological influence in its formation
(Capezzuoli et al., 2014; Perri et al., 2012) and hence, tufa microbialites may be considered a biogenic facies of tufa formed
by a benthic microbial community. As such, these deposits may also be considered part of contemporary tufa-forming
environments and habitats such as 'petrifying springs'. Petrifying spring  and tufa formations have been recognised in a variety
of coastal settings in Europe (EC, 2016; Farr and Graham, 2017; Faulkner and Crae, 2018; Howie and Ealey, 2010, 2009;
Lyons and Kelly, 2016; Rodríguez Guitián et al., 2020).

Research on rock coast microbialites has been approached in numerous ways, including as 'classical' and 'non-classical'
microbialites, as terrestrial carbonate, predominantly tufa, and as part of a wider ecological system or habitat. Each of these
approaches is described and discussed below.

## 2.1 Defining microbialites and microbial mats

Microbial mats are stratified, highly organised, and diverse microbial communities that form a biofilm defined as "*discrete
benthic structures constructed by microorganisms (eukaryotic and prokaryotic; photosynthetic and nonphotosynthetic)*"
(Bauld, 1981; in Grey and Awramik, 2020). Grey and Awramik (2020) state that the term microbial mat should only be applied
to living mats. They are found in a variety of settings including hypersaline, coastal and intertidal, oligotrophic, extreme low
temperature, hot spring and acid environments (Prieto-Barajas et al., 2018). Dependent on the benthic microbial community
or biocenosis, microbial mats may be termed 'algal' or 'cyanobacterial' mats if the microbial mat's primary constituent is
eukaryotic (algal) or cyanobacterial respectively (Bauld, 1981; in Grey and Awramik, 2020); however, use of the term 'algal'
should be avoided due to the lack of a generally accepted definition of this informal term.

Microbial mats are the principal constructional layer of microbialites, including stromatolites and microbially induced
sedimentary structures (MISS) (Noffke et al., 2001; Noffke and Awramik, 2013). MISS are defined as "*sedimentary structures
in siliciclastic sediments and rocks induced by microbial activity*" (Noffke et al., 1996). They occur in siliciclastic, evaporitic,
and carbonate settings (Bose and Chafetz, 2012; Noffke and Awramik, 2013) and have been described globally, notably in the
North Sea (Gerdes et al., 2000), the Mediterranean (Aref, 1998; Gerdes et al., 2000; Lakhdar et al., 2020; Sanchez-Cabeza et
al., 1999), the north-west Atlantic (Cameron et al., 1985), and the Gulf of Mexico (Bose and Chafetz, 2009). They form due
to the trapping, baffling, binding and biostabilisation of detrital sediment by a biofilm or microbial mat and the interaction
with the extrinsic environment (Noffke, 2010, 2008; Noffke and Awramik, 2013). The formation of MISS contrasts with other
microbialites, including stromatolites, which require repeated primary mineral precipitation in the microbial mat's extracellular
polymeric substances (EPS) to accrete into a layered structure (Noffke, 2010; Noffke and Awramik, 2013). Regarding rock

coast microbialites, microbial mats have been recorded from coastal environments, predominantly from the intertidal zone (Gerdes and Krumbein, 1994; Prieto-Barajas et al., 2018). Microbial mats in rock coast settings have been described from Northern Ireland and South Africa as thin mats that may develop on sub-horizontal surfaces in shallow water, often in discharge apron settings (Cooper et al., 2013; Edwards et al., 2017). While the repeated accretion of microbial mats may result in the

formation of tufa stromatolites, the interaction of microbial mats with the rock coast setting may also result in microbial mat features such as 'blister mats' described by Edwards et al. (2017). Thin mats with limited repeat layering or mineralisation share many similarities with MISS due to their comparable interactions with the environment. However, despite their ubiquity, there has been a general lack of description of such microbial mat features.

The repeated trapping and binding of detrital sediment and/or the mineral precipitation by the benthic microbial community may form a stromatolite through repeated stacking of microbial mats (Logan et al., 1964; Noffke and Awramik, 2013). Stromatolites are "*a laminated organosedimentary structure produced by precipitation, or by sediment trapping and binding, as a result of the growth, behaviour, and metabolic activity of micro-organisms, principally cyanobacteria*" (Grey and Awramik, 2020) based upon definitions provided by Awramik and Margulis in Walter (1976). They are considered a subset

of 'microbialites', a term first applied by Burne and Moore (1987) as an umbrella term for stromatolites, thrombolites, dendrolites, leiolites and MISS. These subset forms of microbialites arise through different primary depositional or secondary post depositional and diagenetic processes, and sometimes the nature of the structural drivers is obscure. For example, thrombolites might develop because of distinct microbial communities that give rise to clotted structure, or through bioturbation that destroys primary lamination (Burne and Moore, 1987; Harwood Theisen and Sumner, 2016; Moore and

Burne, 1994; Shapiro, 2000).

'Microbialite' has also been used as a more generic term referring to structures that appear to be microbial in genesis, regardless of their internal structure which may not be visible or initially apparent, or might vary at any given location, as suggested by Suosaari et al. (2019b). The term 'microbialite' was first applied on rock coasts by Smith et al. (2011) and subsequently a

variety of mesostructures has been described. Rock coast microbialites may be considered to cover a range of deposits including thin microbial mats and thicker repeated units forming stromatolites and other microbialites that accrete in a rock coast setting.

## 2.2 Defining tufa and terrestrial carbonates

### 2.2.1 Terrestrial carbonate

Terrestrial (non-marine/continental) carbonate deposits accumulate in a variety of depositional settings including lacustrine, palustrine, cave, fluvial, subaerial, spring, and hypogean environments (Della Porta, 2015). As a result a spectrum of carbonate deposits can develop, including pedogenic carbonates, palaeosols, and caliche/calcretes; palustrine carbonates; cave




carbonates, speleothems and karst; eolian carbonates; glacial carbonates; travertine, calcareous tufa and sinter; lacustrine and fluvial carbonates (Flügel, 2010). In addition, carbonates also accumulate in transitional marine environments including beach

(foreshore), barriers and coastal lagoons and peritidal environments (Flügel, 2010). Within the study of terrestrial carbonates, there is no universally accepted classification (Della Porta, 2015).

### 2.2.2 Tufa

Tufa, a terrestrial carbonate deposit, has an often controversial and complex nomenclature, with core definitions frequently contested, most notably the distinction between tufa and travertine (Capezzuoli et al., 2014; Della Porta, 2015; Ford and Pedley,

1996; Pentecost, 2005). While definitions based upon carbon dioxide source and the chemical mechanism of precipitation have been proposed for tufa (see Capezzuoli et al., 2014), it is often defined as "*the products of calcium carbonate precipitation under cool, ambient temperature freshwater: they typically contain remains of micro-and macrophytes, invertebrates and bacteria*" (Ford and Pedley, 1996; Pedley, 1990) as described by Della Porta (2015). This distinguishes between tufa (meteogene travertine) and travertine *sensu stricto* (thermogene travertine), both constituents of travertine *sensu lato*. If rock

coast microbialites are viewed as a biogenic facies of travertine *s.l.,* the term tufa is best applied due to the ambient temperature of the source springs of microbialite occurrences (Cooper et al., 2013; Dodd et al., 2018; Forbes et al., 2010; Perissinotto et al., 2014; Rishworth et al., 2017, 2016). Furthermore, the cool water temperature definition for tufa often refers to, or requires, cyanobacteria, heterotrophic bacteria and algae to be present and dependent on ambient temperatures <30ºC (Capezzuoli et al., 2014), supporting microbialite facies. It is worth noting that while a temperature-based definition is frequently applied to

tufa, there are issues determining the depositional temperature of the carbon dioxide source of inactive or fossil deposits (Jones and Renaut, 2010; Pentecost et al., 2011).

The presence of biofilms is ubiquitous in terrestrial carbonate and tufa deposits, as demonstrated by Della Porta (2015) and increasingly tufa is considered to be a microbially induced/ influenced sedimentary deposit (Capezzuoli et al., 2014; Perri et

al., 2012). This position does not exclude an abiotic control on calcium carbonate precipitation (Ford and Pedley, 1996; Merz-Preiß and Riding, 1999) but suggests that biofilms may have a substantial or dominant role in the formation of tufa. This has been observed in the field with investigation suggesting that tufa precipitation frequently requires a biofilm, and without the presence of biofilm no carbonate precipitation occurs (e.g., Manzo et al., 2012), even in waters supersaturated with calcium carbonate (Shiraishi et al., 2008). In addition, field experiments with fluvial tufa (e.g., Gradziński, 2010) and in vitro mesocosm

experiments (e.g., Pedley et al., 2009; Rogerson et al., 2010) have demonstrated that biofilms control carbonate precipitation through the presence of EPS, and are required for efficient development of tufa. While it may be tempting to classify any microbially influenced tufa as microbialites, it has been the position that while terrestrial carbonates such as lake crusts and laminated travertines *s.l.* are frequently cited as stromatolites, the use of the term has been applied too loosely (Pentecost and Viles, 1994).



## 2.3 Integration: tufa microbialites or microbial tufa facies?

The role of biofilms and microbial mats in the formation of tufa is increasingly considered to be crucial in their formation. This is also evident in the numerous descriptions of stromatolitic tufa facies within the terrestrial carbonate and tufa focused literature such as fluvio-lacustrine, fluvial and lacustrine tufa deposits (e.g., Arenas et al., 2000, 2019, 2010; Guo and Chafetz, 2012; Manzo et al., 2012; Pedley et al., 1996; Ritter et al., 2017; Rodríguez-Berriguete, 2020; Valero Garcés et al., 2008). This is also extended to thermal travertine s.s., spring deposits (e.g., Gandin and Capezzuoli, 2014). These facies may be considered to be 'tufa stromatolites', a subset of tufa microbialites, bridging the gap between tufa and microbialites and bringing stromatolitic tufa facies under the umbrella of stromatolites and microbialites.

Tufa microbialite, "*a tufa forming as a result of microbial activity (that may be)… laminated, clotted or shrubby*" (Grey and Awramik, 2020) describes a variety of microbially induced deposits, but is most commonly applied to tufa stromatolites. Introduced by Riding (1991), it describes stromatolites "*dominated by the precipitation of minerals on (as opposed to within) organic substrates*" predominantly by cyanobacteria in freshwater lakes and streams. They are distinguished from other forms of tufa such as moss tufa by their lamination. The tufa thrombolite description was subsequently applied by Riding (2000) to describe tufa microbialites with clotted fabrics. Riding (1991) notes that tufa stromatolites can grade into skeletal stromatolites, that preserve the stromatolite-building organisms as calcified fossils.

There are a variety of superseded and alternative terms for tufa microbialites including:

- Microbial tufa: "*formed when microorganic material is incorporated during inorganic carbonate precipitation*" (Burne and Moore, 1987);
- Algal tufa: an algal boundstone originally described from Sleaford Mere, Southern Australia (Warren, 1982);
- Cryptalgal tufa: a term applied by Monty (1976), for microbialites in Green Lake, New York, to describe a "*a tufa inferred to have formed through microbial activity*" (Grey and Awramik, 2020); and
- Cryptomicrobial tufa: "*a tufa that is inferred to have formed through microbial activity*" (Grey and Awramik, 2020).
-

Tufa and tufa microbialites have been excluded from much 'classical' microbialite work, including guides to their description and classification due to a lack of detailed morphological description (Grey and Awramik, 2020). However, the integration of tufa, along with travertines, speleothems, sinter, and microbial crusts within microbialite classification has been advocated (Grey and Awramik, 2020).

Lithologically, *in situ* tufa microbialites (excluding mobile oncoids and detrital tufa microbialites) can be considered, and have been described as autochthonous tufa deposits as they are "*phytohermal constructions where there is an in situ organic framework*" (Ford and Pedley, 1996). As a tufa facies, stromatolitic tufas have been described and classified in multiple studies,





such as phytoherm boundstones as defined by Pedley (1990), as "*in situ stromatolitic build-ups with fringe-cements and often associated with oncoids*" (Pedley, 1990; in Pentecost, 2005).


While the biogenicity of rock coast tufa microbialites has been demonstrated in South African occurrences (e.g., Perissinotto et al., 2014; Smith et al., 2005; Smith and Uken, 2003), the role of microbiota in the formation of other rock coast microbialite localities has not been thoroughly assessed such as those in Co. Sligo, Ireland (e.g., Cooper et al., 2022). For such deposits that appear to be microbial in origin, the terms 'cryptomicrobialite' or 'cryptomicrobial tufa' have been proposed (Grey and

Awramik, 2020). A greater understanding of the role of microbiota and microbialite builders in global occurrences of rock coast microbialites is required to establish biogencity. It would also allow for this distinct facies to be distinguished from other associated terrestrial carbonate facies. Rock coast microbialites are often associated with other non-biogenic tufas (e.g., bryophyte tufas): both surface and hypogean; speleothem, pedogenic carbonates (e.g., paludal carbonates and calcrete) and beachrock and carbonate cements. Associated non-microbialite terrestrial carbonate facies have been described in association

with rock coast microbialites including rhizoliths described by Edwards et al. (2017) and surface-cemented rudites, also called beachrock (Edwards et al., 2017), shell conglomerate (Perissinotto et al., 2014), and sand beach deposits (Cooper et al., 2022). More detailed study of biogenicity and physico-chemical and biological controls over carbonate precipitation will allow for the refinement of the relationships between transitional carbonates such as where cave resurgence associated speleothem grades into tufa microbialite (Pedley and Rogerson, 2010).

**2.4 Habitat and wider ecological implications**

The occurrence of active tufa deposits forms a distinct and ecologically unique habitat. Within Europe, the habitat type 'H7220 Petrifying springs with tufa formation (*Cratoneurion*)' is recognised by the Habitats Directive (Council Directive 92/43/EEC on the Conservation of natural habitats and of wild fauna and flora) as "*hard water springs with active formation of travertine or tufa. These formations are found in such diverse environments as forests or open countryside. They are generally small*

*(point or linear formations) and dominated by bryophytes (Cratoneurion commutati).*" (EC, 2013). This corresponds to the UK National Vegetation Classification of 'M37 *Cratoneuron commutatum-Festuca rubra* spring community' and the 'M38 *Cratoneuron commutatum-Carex nigra* spring community' (EC, 2013; Rodwell, 1992). In Ireland, Lyons and Kelly (2016), define eight petrifying spring plant communities including the group 1 plant community '*Eucladium verticillatum- Pellia endiviifolia* Tufa Cascades'. This plant community describes often near-vertical, steep, tufa cascades that occur on coastal

rocky cliffs within the supralittoral spray zone (Lyons and Kelly, 2016). While Lyons and Kelly (2016) do not systematically survey the microbial fauna of these coastal tufa occurrences, a red alga *Chroothece* sp., was recorded at numerous coastal occurrences and cyanobacteria *Rivularia biasolettiana* and Xanthophyte *Vaucheria* spp. at other inland occurrences. The recognition of rock coast microbialites as part of these habitats may lead to their future identification at other sites. Furthermore, Lismore, Argyll, Scotland has ubiquitous terrestrial carbonate deposits (Faulkner and Crae, 2018), which appear near-identical

in setting and morphology to other British and Irish occurrences of rock coast microbialites.



Other attempts to map global occurrences of rock coast microbialites with greater recognition have been undertaken with the aim to acknowledge and protect their biodiversity. South African occurrences have been recognised in the National Biodiversity Assessment of South Africa (Rishworth et al., 2019) and in south-western Western Australia as 'Augusta

Microbial Threatened Ecological Communities' (Forbes et al., 2010; Onton et al., 2009). One key component to the successful continued presence of these unique habitats is the maintenance of their eco-hydrological habitat requirements. These active tufa deposits can serve as sensitive indicators of water quantity or quality changes due to either direct anthropogenic modification or climate change which may upset the intricate hydrological balance within groundwater dependent rock coast fringe habitats.

**3. Classifying and describing rock coast microbialites**

The following section reviews the numerous classification schemes that may be applied to rock coast microbialites from a number of different research areas including 'classical' microbialites, 'non-classical' microbialites and terrestrial carbonates.

**3.1 Classification and description with a 'classical' microbialite view**

Microbialite classification has been much debated and divided, with application of a great variety in terminology and schemes

across the globe. The following briefly summarises these classification schemes. Grey and Awramik (2020) provide a detailed history of microbialite classification and nomenclature.

The earliest naming of stromatolites involving application of binomial Linnaean nomenclature with the upper Cambrian age Cryptozoön (Hall, 1883) has been documented from New York State, USA (Grey and Awramik, 2020; Lee and Riding, 2021).

Since then, a wide variety of additional classification schemes (eg., Cao and Bian, 1985; Donaldson, 1963; Szulczewski, 1968) have been proposed; including descriptive polynomial schemes suggested by Maslov (1960, 1953) and Hofmann (1969) which were subsequently abandoned due to "*unwieldy terminolog*(ies)" (Grey and Awramik, 2020).

One of the most widely cited classifications was proposed by Logan et al. (1964) who used a descriptive geometric formulae

which abandoned the binomial naming of stromatolites based upon a perceived lack of morphological consistency of biological species. Logan et al. (1964) describes hemispheroids and spheroids (oncolites) as basic geometric units that can be arranged into complex geometric arrangements, e.g. laterally linked hemispheroids (LLH), discrete, vertically stacked hemispheroids (SH), and spheroidal structures (SS). However, this classification system has not been widely adopted, due to its inability to describe more complex microbialite forms (Grey and Awramik, 2020). A vast array of naming classifications and systems

have been applied to stromatolites and microbialites, resulting in the current state of classification being undoubtedly messy and confusing (Grey and Awramik, 2020).



Microbialites, by definition, can be classified by their builders, depositional processes and their products. Living microbial mats can be classified based upon the microbial mat/stromatolite builders. For example, Freytet and Verrecchia (1998) identify algae that form stromatolites through biocrystalization and classify biocenoses based on their ecology (e.g., "*Aquatic environments that are permenantly or periodically immersed… form*(ing) *coatings on the bottom and side of dams, coatings on living or dead organic debris and oncolites corresponding to the Phormidium incrustatum community originally described by Fritsch and Pantin* (1946)*."*). In some cases, tufa stromatolites have been classified by a dominant builder (e.g., *Schizothrix* tufa of Irion and Müller (1968)). These depositional processes (e.g., biologically influenced/biochemical calcification) are intrinsically linked to their products and this has lead to a number of attempts to genetically classify microbialites (e.g., Burne and Moore, 1987; Riding, 1991).

In addition to 'classical' microbialites and an ecological view of microbial mats, numerous authors have considered the classification of the features of microbial mats and MISS with  genetic schemes for MISS by Gerdes et al. (2000), Noffke et al. (2001), Schieber (2004) and (Eriksson et al., 2007a).

The description of microbialites is most commonly aided through the use of the traditional heirarchial scale of observation from mega-, macro-, meso- to micro-structure as defined by Shapiro (2005) and discussed by Grey and Aramik (2020) (Fig. 1a). These levels are not mutually exclusive and are partially open-ended (Grey and Awramik, 2020). Other spatio-temporal scales have been suggested. For example Ibarra and Corsetti (2016) define scales of control as either local, a process at the scale of the stromatolitic structure, at the metre to micrometre scale, (e.g., crystal growth; biofilm trapping and binding; post-depositional dissolution); or non-local, refering to processes greater than the stromatolitic structure at the kilometre or larger scale (e.g., seasonality; post-depositional burial and deformation).

## 3.2 Classification and description with a terrestrial carbonate view

Due to the morphological and geochemical similarities at the macroscale, rock coast microbialite classifications have also been closely aligned to other terrestrial carbonate deposits. This is despite a current lack of a universal nomenclature for terrestrial carbonates (Della Porta, 2015). For example, calcite and aragonite speleothems may be considered hypogean travertines (Pentecost, 2005), although the classification of speleothems is distinct from other terrestrial carbonates, with a vast array of types (see Hill and Forti (1997)). Tufa can be classified through a variety of non-mutually exclusive criteria, commonly through geochemical, fabric and morphological criteria (see Pentecost, 2005; Pentecost and Viles, 1994) and these classification schemes are briefly summarised here.

Recognised by Pentecost (1993), tufa can be classified based upon geochemistry, and the origin of the carrier carbon dioxide has been used to distinguish between travertine *s.s.* and tufa. The origin of the carrier affects the isotopic composition of the



deposit and influences the morphology and fabric (Pentecost, 2005). Pentecost (1993) considers meteogene travertines (classified here as tufa) deposited from a meteoric water source and thermogene travertine (travertine *s.s.*) deposited from a thermal water source (Pentecost, 1993; Pentecost and Viles, 1994). Tufa fabric, "*the architecture of the deposit (i.e., the arrangement, density and size of the building units)*" (Pentecost and Viles, 1994) can also be classified, often with a focus on the biological influences. It is present at the micro- and meso-scale, referred to as the micro- and meso-fabrics,  with the former

relating to fabrics identifiable at the microscopic level through the use of thin section analysis and SEM, and the latter refering to fabrics visible by eye in a hand specimen (Pentecost, 2005). One of the most commonly applied classifications is that of tufa morphology or geomorphology. A seminal classification corresponding to environmental setting by Pedley (1990) has been adapted by Pentecost (1993) and Pentecost and Viles (1994) and was further expanded upon by Ford and Pedley (1996). This thread of geomorphological classifications has been frequently applied to British and Irish tufa deposits (e.g., Farr and

Graham, 2017; Lyons and Kelly, 2016), partially due to its British and north west European scope (Pentecost, 1993); occasionally, they have also been applied globally (Forbes et al., 2010).

Tufa systems are commonly described and classified through facies approaches that can integrate the above classifications. One of the most frequently applied is that of Ford and Pedley (1996) who divide tufas into allochthonous and autochthonous

tufa deposits, the latter of which is subclassified into three facies (Jones and Renaut, 2010), based upon established carbonate classification schemes.

- 'Boundstone sheets of micrite and peloids (stromatolith-like and bacterioherms)'
- 'Microherm shrubby framework of bacterial colonies'
- 'Framework true "reef" framework of macrophytic coated with mixed microtic and sparry calcite fringes'


Comparable to the hierarchical mega- to micro-structure scale applied to microbialites, similar classifications have been proposed for terrestrial carbonates. For example, Ordonez et al.*,* (1986)*,* considered a hierarchical scale describing tufas through macrostructure, controlled by the relative position of the water table to topography; mesostructure, controlled by the ecology and encrustation nuclei, and microstructure, controlled by the efficiency and type of degassing. Guo and Riding (1998)

also applied a three-part classification scheme to a Late Pleistocene travertine system, considering lithotypes (e.g., reed travertine, shrub travertine), depositional systems (e.g., mound depositional system, slope depositional system), and facies (e.g., waterfall facies, terrace slope facies).

### 3.3 Integrated classification and towards a rock coast microbialite classification

As with tufas, rock coast microbialites may be classified based upon their geochemistry and fabric. However, as a microbialite

facies of tufa that occurs in a specific setting, it can be considered, given the current investigations, to be of a relatively consistent geochemistry and fabric. Rock coast microbialites are partially defined by their characteristic carrier carbon dioxide source of supratidal feshwater springs emerging from a carbonate saturated groundwater source (Rishworth et al., 2020). This



results in the bulk mineralogy for rock coast microbialites dominated by calcite and aragonite (Dodd, 2019; Forbes et al., 2010; Rishworth et al., 2020). As fabric is examined, rock coast microbialites are defined as a tufa microbialite facies with distinct

fabrics (i.e., clotted thrombolites, laminated stromatolites and aphanitic leiolites). While the classification of geochemistry and fabric may be suitable to apply to rock coast microbialite occurences, a geomorphological classification of rock coast microbialites is frequently applied and is equated to microbialite macrostructure (e.g., barrages). Recently, a facies approach to rock coast microbialites has been applied by Cooper et al., (2022), which considers a general model of microbialite facies on siliclastic rock coasts. A similar approach was undertaken by Forbes et al., (2010).


Trompette (1982) proposed a morphogenetic model, suggesting that the environment was the dominant influence on stromatolite formation at the macroscale, and that the biological influences predominated at the microscale (Suosaari et al., 2019b; Trompette, 1982) (Fig. 1b). This model was modified by Suosaari et al., (2019b) on examination of microbialite systems in Hamelin Pool and the Bahamas. These hierachal structural models for both tufas and microbialites demonstrate comparable

controls on morphology, and subsequently may be applied to tufa microbialites; however, such models would need to be tested. Therefore, the approach taken here is that of a hierachal scale that equates to microbialite mega-, macro-, meso- and micro-structure with the application of terminology from terrestrial carbonates. While this appears most suitable at the macro-scale where environmental influcene is greatest and microbialite terminology may also be applied at the meso- and micro-scale (Fig. 1b-c).





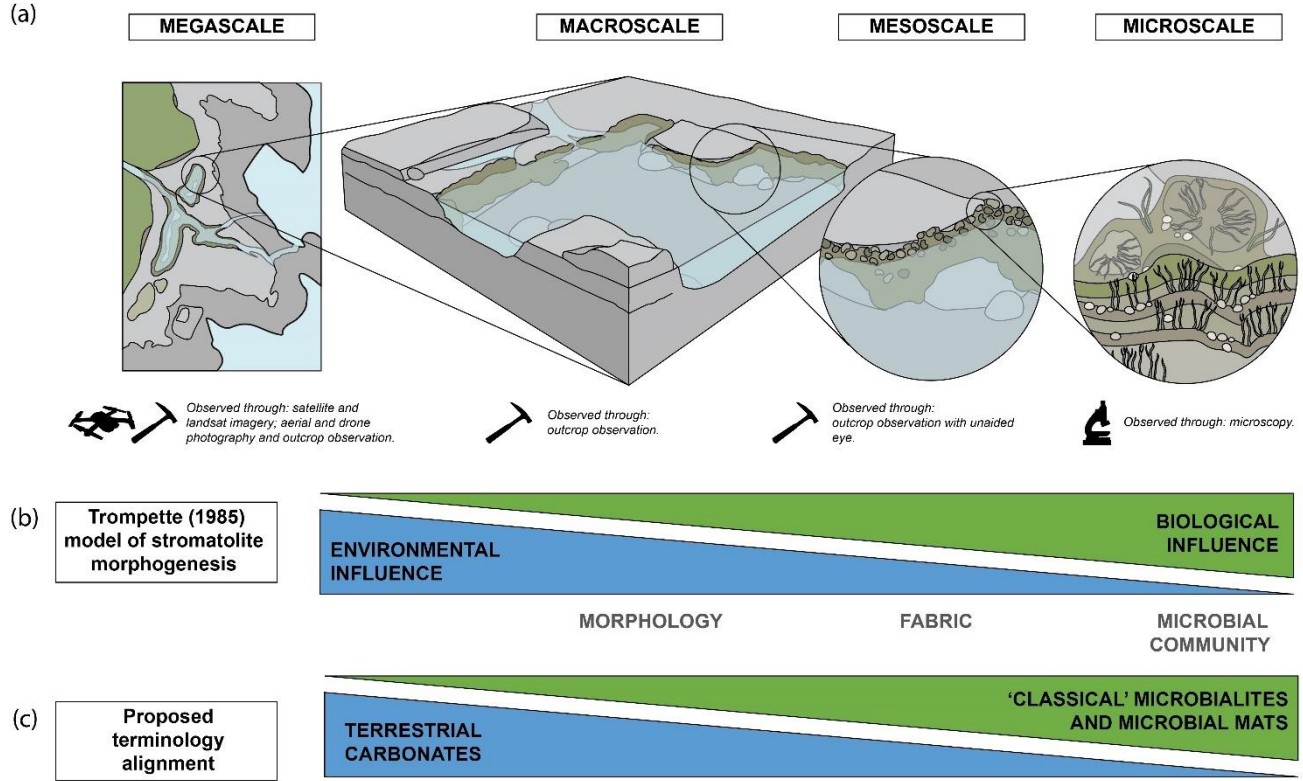


**Figure 1: (a) Diagram of hierachial structural scale from megascale to microscale (based on Grey and Awramik, 2020; Shapiro, 2005); (b) Trompette (1982) model of stromatolite morphogenesis, adapted from Suosaari et al., (2019b); (c) the proposed terminology alignment between terrestrial carbonates and 'classical' microbialites and microbial mats.**

## 4 An integrated geomorphological classification for rock coast microbialites

As discussed, the terminology used for rock coast microbialites is somewhat divided between 'classical' microbialite and terrestrial carbonate terminology. In combination with this, the global distribution of rock coast microbialites, despite the majority of research output being based upon localities in South Africa (Rishworth et al., 2020), has lead to inconsistencies in the terminology applied, and the need for consolidation, unification and definition. Thus, an integrated rock coast microbialite classification is required for future research of rock coast microbialite systems and, importantly, the comparision with

terrestrial and marine carbonates. The proposed classification system will address the presence of synonyms, duplications, omissions, and the wide variety of terms applied to the same morphological features, predominantly at the macro- and meso-scale. This is approached using the hierarchical microbialite scale as defined by Shapiro (2005) and discussed by Grey and Awramik (2020), with discussion on the strengths and weaknesses of terms and their source classification schemes.



While a rock coast microbialite classification system is considered to be important for future research, care must be taken to avoid excessive catergorisation, especially upon morphogenesis which remains poorly understood in rock coast microbialites. Current understanding of terrestrial carbonate morphogenesis is already much more advanced and may be comparable at macroscale, yet the relationships between these deposits are not understood well enough for direct correlation. While terrestrial carbonates can already be classified based upon fabric and geochemistry, the current lack of description and suficiently large

datasets concerning rock coast microbialites, currently excludes such classification schemes.

### 4.1 Megastructure

Megastructure addresses the largest- (meter to decimeter) scale aspects of microbialite occurrence and their respective beds, examining the bed-scale or stratum-scale structures, including buildups such as the largest bioherms or biostromes (Grey and Awramik, 2020; Shapiro, 2005). This may refer to the location (e.g., aerial extent and stratigraphic setting); surrounding strata

relationships (substrate, initiation, interface and growth direction); mode of occurrence (e.g., buildup, bioherm and biostrome) and shape (Grey and Awramik, 2020). For the study of rock coast microbialites, the megastructure has been commonly described within the study site setting.

### 4.2 Macrostructure

Following Grey and Awramik (2020), macrostructure refers to aspects of the majority of bioherms and biostromes, including

features of microbialite gross morphology, and describes features such as shape (e.g., stratiform, oncoidal, and columnar), plan view, and linkage.

Rock coast microbialite macrostructure has followed the definition for microbialite macrostructure relatively closely (Edwards et al., 2017); however, there is generally a focus on the gross morphology of microbialite occurrences opposed to shape (e.g.,

columnar), which is generally considered a feature of mesostructure. Terminology has also frequently been adapted from terrestrial carbonate nomenclature (Fig. 2). For example, Forbes et al., (2010) and Edwards et al., (2017) presented depositional models for coastal tufa microbialites following the classification scheme of Ford and Pedley (1996). In comparable terrestrial carbonates, the classification is frequently based upon criteria like botanical content, depositional processes, geochemistry, fabric, facies and morphology which produces a number of different schemes (Fig. 2). Pedley (1990), Ford and Pedley (1996)

and Pentecost and Viles (1994) provide comparable morphology and environmental setting- based classifications, broadly equivalent to tufa macrostructure proposed by Ordonez et al., (1986) and considered to be controlled by the position relative to the water table and topography.






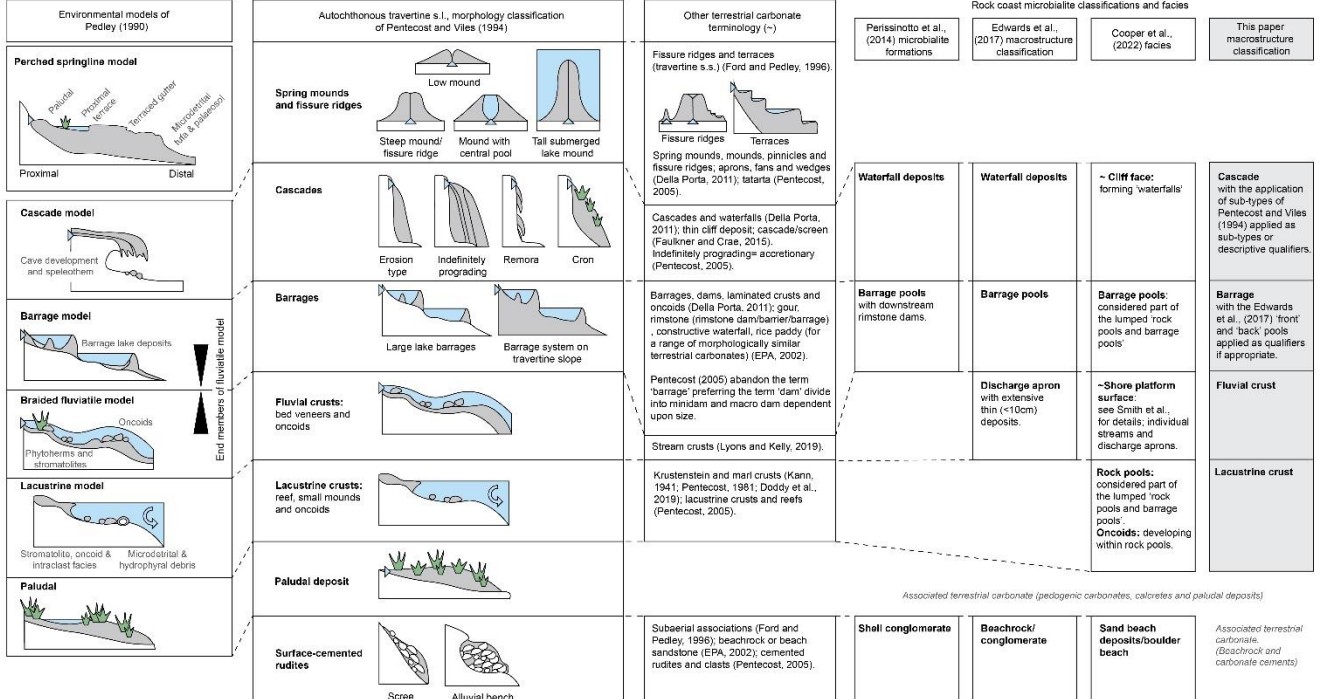

**Figure 2. Comparison of terrestrial carbonate terminology (applying key classification schemes of Pedley (1990), Pentecost and Viles (1994) and other terminology (Della Porta, 2011; Faulkner and Crae, 2018; Ford and Pedley, 1996; Lyons and Kelly, 2016; Pentecost, 2005; Field, 2002)) and rock coast microbialite terminology (Cooper et al., 2022; Edwards et al., 2017; Perissinotto et al., 2014)**

Following the adaptation of tufa nomenclature by Forbes et al. (2010) to produce two depositional models featuring the perched

springline model, fluvial barrage model and cascade of Ford and Pedley (1996), Perissinotto et al. (2014) considers 'stromatolite formations' consisting of barrage pools with downstream rimstone dams, waterfall deposits (sensu Forbes et al. (2010)) and associated 'shelly conglomerates'. This seminal work strongly influenced subsequent studies such as Edwards et al. (2017) who along with the previous detailed macrostructures also included discharge aprons. The shore platform setting (Smith et al., 2018), contains microbialites composed of thin crusts, barrage pools, and shallow rock pools. This shore platform

setting was considered in a microbialite 'modes of occurrence' facies approach by Cooper et al. (2022) alongside cliff face, boulder beach, rock pool and barrage pools, oncoids and sand beach deposits. The consistent inclusion of surface cemented rudites (shell conglomerate, beachrock, sand beach deposits etc.,) without direct evidence of biogenicity means that they are not included within the current classification, however, that is not to say they should not be included within wider study of rock coast terrestrial carbonates (Fig. 3).





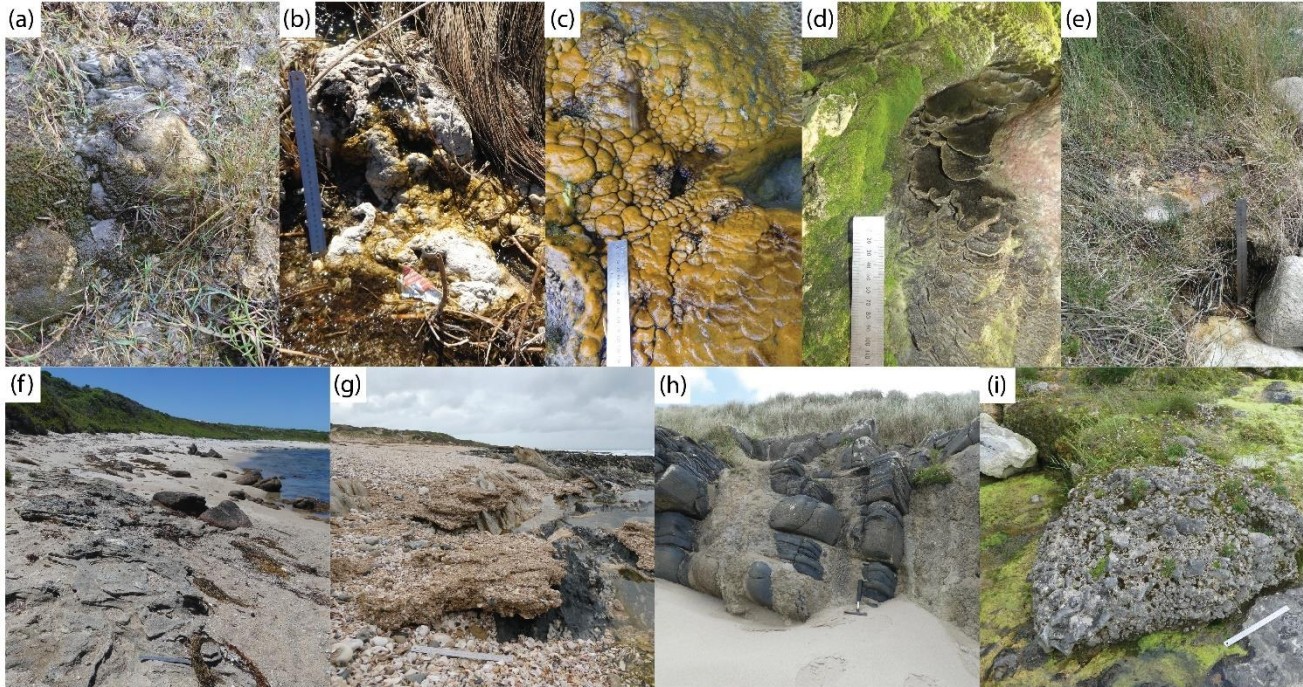


**Figure 3. Terrestrial carbonate structures commonly associated with rock coast microbialites: (a) bryophyte tufa cushions (Ardglass, Northern Ireland); (b) surface (non-microbialite) tufa (Schoenmakerskop, South Africa); (c) hyopgean tufa (Quarry Bay, Western Australia); (d) Speleothem (Mullaghmore, Ireland); (e) paludal tufa (Canal Rocks, Western Australia); (f)-(i) incipient beachrocks and carbonate cements: (f) Quarry Bay, Western Australia; (g) Lauries Bay, South Africa; (h) Saligo Bay, Isle of Islay, Scotland; (i)**
**Sligo Bay; Ireland).**

The following geomorphological classification distinguishes between the geomorphology of the microbialite deposit and the setting or mode of occurrence (e.g., fluvial and lacustrine crust macrostructures present in a discharge apron or shore platform setting). Due to this complex nomenclature, the following terminology is suggested (Fig. 4).



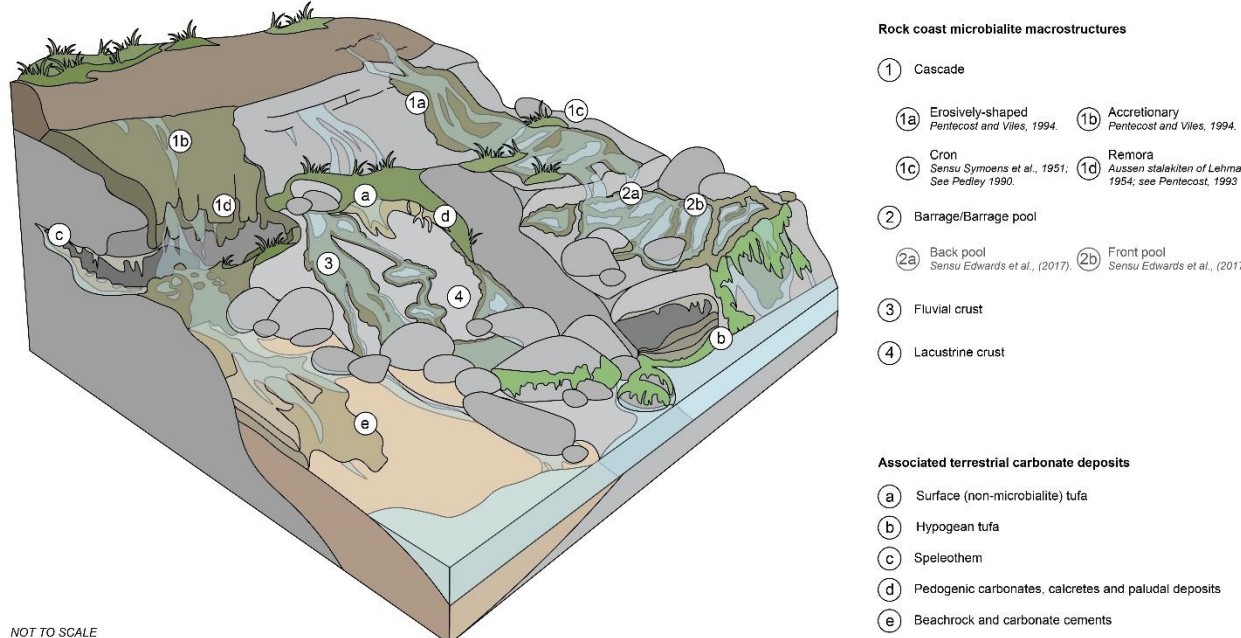

**Figure 4. Block diagram of rock coast microbialite macrostructure and associated terrestrial carbonate deposits.**





**Figure 5. Rock coast microbialite macrostructures: (a) erosively-shaped cascade (Cape St Francis, South Africa); (b) remora (Quarry Bay, Western Australia); (c) accretionary cascades (Aughris Head, Ireland); (d) fluvial crust (Sligo Bay, Ireland); (e) fluvial crust (Sligo Bay, Ireland); (f) barrages (Sligo Bay, Ireland); (g) barrages (Lauries Bay, South Africa); (h) inactive barrages (Quarry Bay, Western Australia); (i) lacustrine crust (Bundoran, Ireland); (j) lacustrine crust (Mullaghmore, Ireland).**

### 4.2.1 Cascade

In rock coast microbialite nomenclature, a variety of terms have been used to describe tufa microbialite deposits on steep gradients; however, cascade deposits (Fig. 5a-c) have been rarely termed as such. Workers have frequently cited the 'waterfall deposit' of Forbes et al., (2010) (Edwards et al., 2017; Perissinotto et al., 2014) despite considering that such deposits bear a striking resemblence to the morphology of tufa deposits (Edwards et al., 2017). The cliff face microbialite facies described by Cooper et al., (2022), is considered to be dominated by the cascade macrostructure, as the cliff face facies is partially defined by its steep shore topography that produces cascade macrostructures.





Pentecost and Viles (1994) considered tufa 'cascades' and identified two distinct forms controlled by variable
erosion/deposition rates. The first form is 'erosively-shaped' deposits, paraboloid in section, with the surface morphology
controlled by spate water trajectory (Fig. 5a) and a second form: 'accretionary' deposits, where the rate of deposition is greater
than the rate of erosion resulting in progradation (Fig. 5c). This second form consists of subtypes. These include keeled
cascades which consist of a narrow slot constricting water flow before falling over a travertine s.l., nose and 'tubes' forming a
travertine spout at the top of cascades (Pentecost, 2005, 1993). 'Cron' is a poorly observed cascade subtype, of small staggered
dams with boggy pools, and an intermediate between cascades and barrage pools (Pedley, 1990; Pentecost, 2005, 1993;
Symoens et al., 1951). Considered a form of speleothem by Pentecost (2005), 'Aussen stalaktiten' (Lehmann, 1954), renamed
remora (Pentecost, 1993), is a stalactite-like mass of terrestrial carbonate that develops on steep slopes in slow or periodic
water flow (Pentecost, 2005) (Fig. 5b). Preferential precipitation of carbonate may result in remora extending from a vertical
surface, potentially due to algal growth (Dobat, 1966; Rong et al., 1996) resulting in a 'phototrophic stalactite' (Pentecost,
2005). Ford and Pedley (1996) also consider waterfall or cascade tufas, but consider that these forms may be constituents of
larger systems and that elements of cascade and paludal models described by Pedley (1990) may be encompassed by the
proximal element of the perched springline (slope) model. In addition, comparable descriptive terms have been applied to
cascade or cascade-like tufa deposits including waterfalls and thin cliff deposits and screens (e.g., Faulkner and Crae, 2018).

In the future, the resumption of the term 'cascade' describing this specific macrostructure is suggested for further work, in
order to agree with terrestrial carbonate and tufa nomenclature with macrostructures such as cron and remora applied as
qualifiers or subtypes.

### 4.2.2 Barrage

In rock coast microbialite nomenclature, barrage deposits (Fig. 5f-h) have been relatively consistently described as such.
Perissionotto et al. (2014) and Edwards et al. (2017) describe barrage pools, with rimstone dams (Edwards et al., (2017)
considers rimstone itself to be a distinct mesostructure). Edwards et al. (2017), also distinguishes between 'back pools' and
'front pools' based upon their position in relation to sea-level and morphology. In relation to morphology, back pools form
due to microbialite growth on both the pool bottom and margins, forming rims that ultimately close the pool; and front pools
form with limited growth at the rims. Edwards et al. (2017) ascribes the morphology of these pools to their level of maturity,
however, this morphogenetic classification is yet to be applied outside of the South Africa. Cooper et al. (2022) also describes
the barrage pool morphology as part of a lumped rock pool and barrage pool facies, combining barrage pool and lacustrine
crust macrostructures. The barrage macrostructure is not mutually exclusive to this depositional facies and may also be present
on shore-platform surfaces and merge with cascade macrostructures in cliff face settings.



Pentecost and Viles (1994) differentiate barrages from cascades based upon vertical accretion of 'barrages' resulting in water impoundment forming 'barrage pools' of which two forms are recognised. The first form are those forming large lake barrages forming on obstructions or breaks in gradient on the accretion surface, and the second are barrage systems that form on preexisting travertine *s.l.* slopes.  Ford and Pedley (1996) describe the barrage model, following the fluvial barrage model of Pedley (1990). Pentecost (2005) re-describes barrages stating that the term barrage suggests a specific obstruction, and applies

the term 'dam' instead. Dams are divided into two groups based upon the the distance between consecutive dams (or inter-dam distance) with dam systems with an inter-dam distance of 1cm - 1m described as minidams and those with inter-dam distances of 1m - 100m termed macrodams. However, this division is purely artificial with no biomodality or discontinuity between forms (Pentecost, 2005). In addition to tufa barrages, similar barrage-like morphologies occur in other terrestrial carbonates such as speleothem deposits where approximate terms 'rimstone' and 'gours' are applied, with 'microgours' used

to refer to morphologies at the cm-scale and travertine *s.s.* and silica sinter where they are termed terraces  (Hammer et al., 2010; Pentecost, 2005). A scaled approach to barrage terminology is also applied by Fouke et al. (2000) and Bargar (1978) to describe travertine morphologies of Yellowstone National Park, USA, who use terraces, terracettes and microterraces.

For further work, the term 'barrage' or 'barrage pool' is suggested in line with both terrestrial carbonate and rock coast

microbialite nomenclature, with constituents of a 'barrage', the outer wall of the structure, backed by a barrage 'pool'. The morphogenetic classification of 'front' and 'back' barrage pools of Edwards et al., (2017)*,* may be applied as qualifiers if appropriate; however, these models should be further tested globally.

### 4.2.3 Fluvial crust and lacustrine crust

Fluvial and lacustrine crusts (Fig. 5d-e, i-j) are often not distinguished in rock coast microbialite nomenclature with both forms

being 'lumped' within a singular setting or facies. Initially, Edwards et al. (2017), described 'discharge aprons' as a distinct macrostructure, describing microbialite growth typically <10cm, on inclined bedrock with flowing freshwater. Inactive and active discharge apron fans are also described by Forbes et al. (2010) from Western Australia. This macrostructure is encompassed by the shore-platform setting described by Smith et al. (2018), with stromatolites accreting on the subhorizontal element of rocky coasts, typically within rock pools of shore platforms.  At the macrostructure level they are simply described

as being thin crusts (1-30cm thick) in addition to low mounds, barrages, and oncoids, as described by Edwards et al. (2017). The shore platform surface facies is also documented by Cooper et al. (2022) to describe rock coast microbialites that develop on the horizontal element of rock coasts, which are thin mats covered by a freshwater film.

Fluvial crusts, also called stream/spring crusts, are characterised by thin crust deposits under flowing water that intergrade

with other macrostructures including cascades and barrages (Pentecost, 2005, 1993; Pentecost and Viles, 1994) as part of the fluviatile model of Pedley (1990). There is recognition that these deposits are frequently associated with cyanobacteria and algae (Pentecost, 2005, 1993). Ford and Pedley (1996) consider low-angle sheet like deposits as part of the distal component





of the perched springline (slope) model and stream bed deposits of cyanobacteria dominated lenses and small bioherms of stromatolite boundstone as part of the braided fluvial model. A comparable speleothem morphology is flowstone, which is

considered analogous by Pentecost (2005); however the surface morphology/texture is smoother due to the lack of macrophyte growth and incorporation of detritus.

Microbial carbonates are considered a distinct lacustrine carbonate facies, identified by MISS and "carbonate biostructures" that include coated grains (e.g., ooids and oncoids), other microbialites (e.g., stromatolites, thrombolites, and other microbial

mats) and tufa mounds (Alonso-Zarza and Tanner, 2010). These are separated from fluvial crusts by Pentecost (2005), who divides lake deposits into lacustrine crusts and lacustrine reefs. The former, lacustrine crusts have been defined similarly to fluvial crusts, consisting of superficial crusts and  oncoids in static bodies of water  (Ford and Pedley, 1996; Pentecost and Viles, 1994). The latter, reef-like deposits are also classified in this lacustrine setting, with acknowledgement that such reefs are commonly laminated and microbially precipitated, i.e., stromatolitic (Ford and Pedley, 1996; Pentecost and Viles, 1994).

This recognition of microbial influence has led to the term 'krustenstein' (Kann, 1941), which has been applied to tufa/marl microbialite crusts (Doddy et al., 2019; Pentecost, 1981).

There is a clear discrepancy between the terminology applied to rock coast tufa microbialite macrostructure and the equivelant tufa morphologies, based upon the unusual setting that rock coast microbialites form, with the setting (e.g., shore platform or

discharge apron) frequently adopted as a macrostructure, reserved here for fluvial and lacustrine crusts. Fluvial crusts should be applied to thin crusts with no water impoundment from barrages with flowing freshwater, while lacustrine crusts should be applied to crusts beneath predominantly no flow in a topographically controlled basin (e.g., a rock pool) (Fig. 5i-j). In reality these two macrostructures may form a gradient covering a range of hydrodynamic conditions on a sub-horizontal surface (e.g., on a shore platform surface) and may grade into other macrostructures such as barrages.

**4.3 Mesostructure**

Mesostructure, according Grey and Awramik (2020), refers to the visible internal organisation and is the hierarchial level that allows for the differentiation of microbialites as stromatolites, thrombolites, dendrolites and leiolites. In living microbial mats, the mesostructure has also been termed 'mat topography' and refers to the surface features as opposed to the visible internal organisation (Bauld, 1992 in; Grey and Awramik, 2020) with 'architecture' proposed as a fossil equivalent (Grey and

Awramik, 2020). The internal organisation of microbialites has been referred to as macrofabric (e.g., Riding, 2011) and mesofabric (e.g., Edwards et al., 2017), however, Grey and Awramik (2020) suggest that fabric is best used in its sedimentological meaning as a constituent of microstructure.

The first mat topography mesostructure of rock coast microbialites was described by Smith and Uken (2003) as coalescing

domal structures, now termed colloform, and following this initial description, Smith et al. (2005) examined the laminated





internal organisation further. The first substantial observation of mesostructure was undertaken by Smith et al. (2011), who described three growth forms: pustular, laminar/columnar and colloform. Described as mesofabrics Perissinotto et al. (2014) also observed these three forms on the South African coast and Cooper et al. (2013), compared these to occurences in Northern Ireland, describing colloform and laminar forms. This was greatly expanded upon by Edwards et al. (2017) who also classified

rimstone, rootcasts, wrinkles, and blister mat mesostructures. Comparably with surface-cemented rudites, or beachrock, the rootcast mesostructure, forming at vegetated cascades resulting in the precipitation of carbonate on roots, is not explicitly biogenic and cannot be included within a classification of rock coast microbialite mesostructure. In the description of the mesostructure of rock coast microbialites, both the growth pattern (visible internal organisation) and the suface morphology (mat topography) have been considered, in some cases mutually exclusively.


The surface morphology of microbial mats and single-taxa microbial colonies can be classified. Whitton et al., (2000) classify the shape of cyanobacterial colonies as: spherical/subspherical, hemispherical, cube, plate, cathrate, floc/amorphous, tuft/bushy, film/mat and crust/fleck. These colonies may form part of the greater surface morphology of microbialites, and demonstrate the biological control on mesostructure; this in combination with the equal environmental control, is the genesis

of mesostructure (Suosaari et al., 2019b; Trompette, 1982). Mesostructure is frequently described and classified on a case-study basis, for example, microbialites at Shark Bay, Western Australia, are classified based upon the surface morphology and degree of lamination. This identified pustular, smooth, colloform, and cerebroid morphologies (Jahnert and Collins, 2012, 2011; Logan et al., 1974; Suosaari et al., 2019a). However, as noted by Jahnert and Collins (2012), classification based on surface morphology may not be useful regarding the fabric and internal morphologies due to complex or compound

microbialites with possible growth hiatuses. The term 'mat topography' has been applied to the surface mesostructure of living microbial mats (Bauld, 1992; Grey and Awramik, 2020). Rock coast microbialite terminology has used a wide variety of terms to describe both the mat topograpy and the internal organisation/laminar profile with terms often being combined (e.g., laminar flat mesostructure of Edwards et al. (2017)); however, care should be taken so as not to consider these two aspects of mesostructure mutually exclusive unless there is sufficient evidence. This may be addressed through the application of the

hierarchical mega- to micro-structure approach. The following classification distinguishes between the mat topography (visible from the exterior) and the internal organisation (which requires examination in section) (Fig. 6).





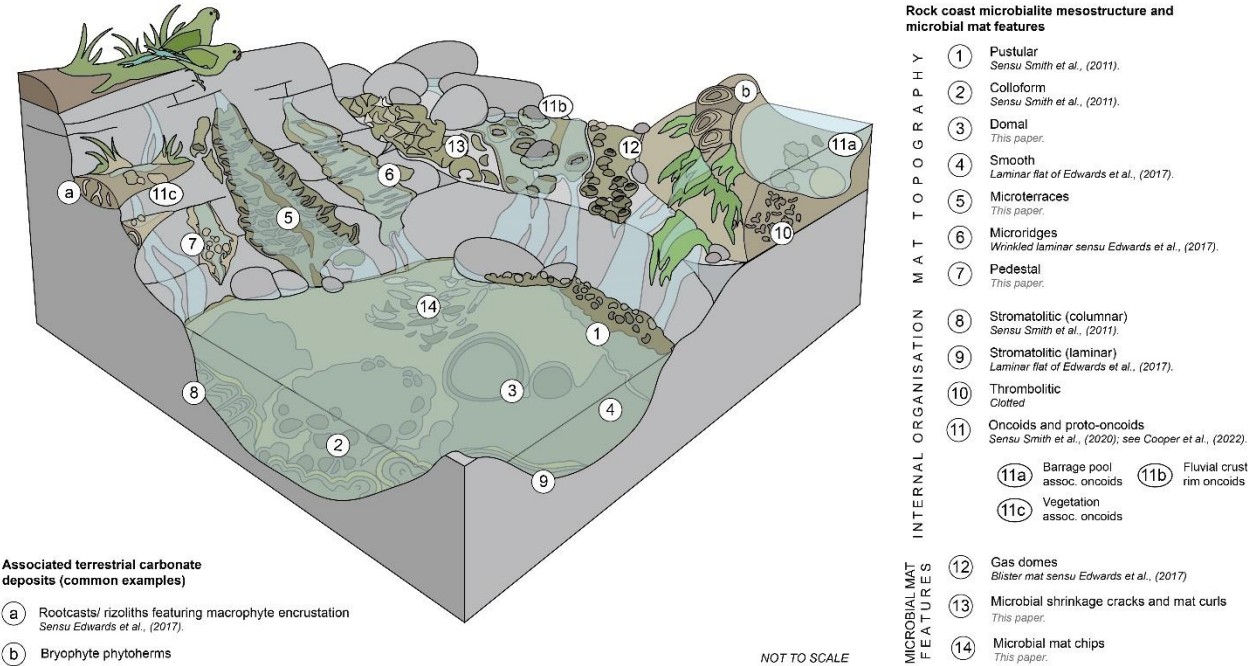

**Figure 6. Block diagram of rock coast microbialite mesostructures including mat topography (surface morphology), internal organisation, and microbial mat features; and associated features of terrestrial carbonate deposits.**

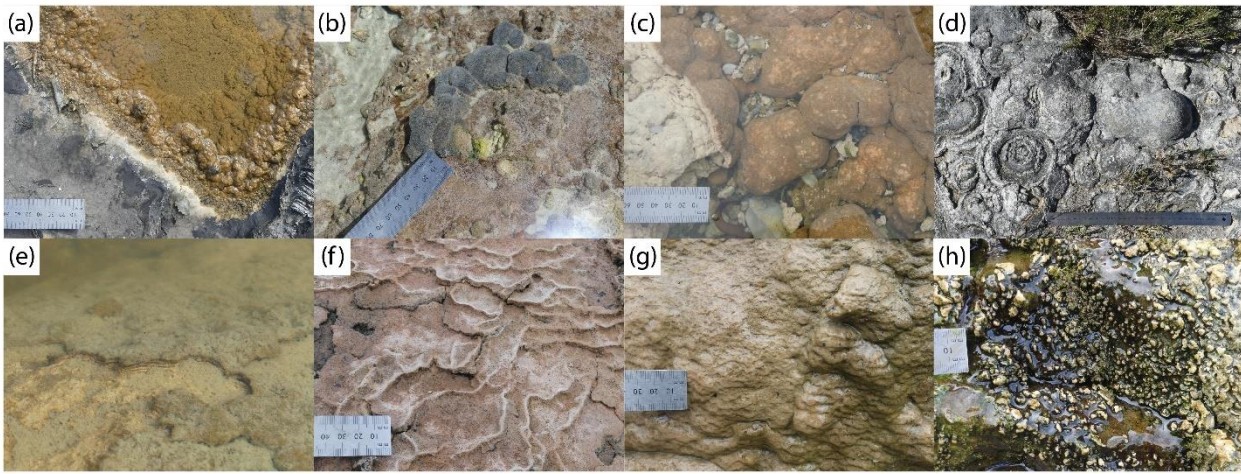


**Figure 7. Rock coast microbialite mat topographies (mesostructure): (a) pustular (Streedagh, Ireland); (b) colloform (Quarry Bay, Western Australia); (c) colloform (Lauries Bay, South Africa); (d) domal (Conto Spring, Western Australia); (e) smooth (Mullaghmore, Ireland); (f) microterraces (Sligo Bay; Ireland); (g) microridges (Bundoran, Ireland); (h) pedestal (Sligo Bay, Ireland).**





**4.3.1 Mat topographies: smooth, pustular, colloform, and domes**

Smooth, pustular and colloform mat topographies have been described from rock coast microbialites and microbial mat and 'classical' microbialite systems, including Shark Bay, Western Australia (Jahnert and Collins, 2012, 2011) and the Exuma Cay and Highborne Cay Bahamas (Reid et al., 1995; Stolz et al., 2009). These terms have also been applied to the external biofilm surface of tufa deposits (e.g., Perri et al., 2012).

A laminar mesostructure was initially identified as LLH-C (see Logan et al., 1964) in the wind-shadow margins of pools by Smith et al. (2011). This 'laminar flat' mesostructure was also identified by Edwards et al., (2017) and Smith et al. (2018) as continous stromatolite growth forming in the wind shadow margin of shallow pools, discharge aprons and ephemerally wet environments (Fig. 7e). It is similar to the 'flat sheet' form of microbial laminates described by Forbes et al. (2010). Smooth mesostructure is associated with a flat laminar internal fabric (Edwards et al., 2017; Smith et al., 2011); however, as with other mat topographies, the relationship with internal structure is not well resolved. Comparable smooth surface morphologies have been described from Shark Bay as "*flat, smooth surfaces with a beige colour either as mats or sub-spherical heads*" (Jahnert and Collins, 2011)*,* with a laminated, stromatolitic fabric and smooth mats in peritidal settings (Gerdes and Krumbein, 1994) and Highborne Cay, Bahamas as smooth mats (Stolz et al., 2009). The term 'smooth' is frequently applied to microbialites and microbial mats and is recommended for further use.

A pustular mesostructure was initially described in the rock coast setting by Smith et al. (2011) from partially emergent settings surrounding pool rims and shallow water and has subsequently been identified frequently (Edwards et al., 2017; Smith et al., 2018) (Fig. 7a). Edwards et al., (2017) further define the pustular mesostructure as being formed of "*small (0.5-2cm wide) irregular shaped nodules which often grow on discharge aprons or at shallow margins of barrage pools*". A comparable pustular mesostructure was first described from Shark Bay, Western Australia by Logan et al, (1974) to describe "*brown surfaces of gelatinous pustules composed of mucilage (1-2cm thick)*" (Jahnert and Collins, 2011) from the shallow intertidal environment. The pustular morphology is described as having a clotted fabric and is therefore thrombolitic (Jahnert and Collins, 2012). It has also been described from normal salinity marine environments in Exuma Cays, associated with *Schizothrix* mats, (Reid et al., 1995) and Highbourne Cays, Bahamas, to describe *Schizothrix* and *Solentia* mats, and biofilms (Stolz et al., 2009); hypersaline *Rivularia*-rich microbial mats of Laguna Negra, Argentinia (Gomez et al., 2018; Mlewski et al., 2018) and peritidal environments (Gerdes and Krumbein, 1994). The term 'pustular' as a mat topography mesostructure is recommended with respect to rock coast microbialites to describe the upper surface of microbial mats.

The colloform mesostructure was also initially described by Smith et al. (2011) from deeper water than the pustular mesostructure to depths of 20-30cm (Fig. 7b-c). The colloform mesostructure has been frequently described (Cooper et al., 2013; Edwards et al., 2017; Smith et al., 2018). Edwards et al. (2017) also considers colloform mesostructure as being depth





controlled, being found on barrage pools and pool walls. It is defined as having "*an interconnected bulbous appearance similar to that of malachite*" (Edwards et al., 2017). This colloform mesostructure is similar to the phytohermal 'bubble form' from
Western Australian microbial laminates described by Forbes et al. (2010). Based upon the pustular and colloform mesostructures at Seaview west, South Africa, Edwards et al. (2017) conclude that a depth controlled continuum exists between the two growth morphologies. The colloform mesostructure is described as "*beige to brown elongate prismatic, spherical and club shaped structures"* with each hemispherical head being 1-5cm in size (Jahnert and Collins, 2011) from subtidal Shark Bay, Western Australia.  The colloform mesostructure is well understood and used within other microbialite literature so is
suitable for further use.

Larger interconnected hemispherical structures with heads up to 15cm in diameter have been observed between  Cape Freycinet and Conto Spring, Western Australia, at an inactive microbialite barrage pool (Fig. 7d). While their morphology is comparable to colloform, the structures are much larger and could be considered a distinct mesostructure/morphology. Indeed as with
pustular and colloform, it is likely that a continuum extists between these mesostructures. The term' domal' is suggested although this does not refer to mat features such as gas domes, petees and mat expansion structures forming from gas accumulation  (e.g., microbial mat decay (Bouougri et al., 2007)).

**4.3.2 Mat topographies: microterrace, microridge and pedestal**

Edwards et al. (2017) described 'wrinkles' as "*small (1-5cm wide) drooping layers*" forming on a sub-vertical faces that merge
with a laminar flat mesostructure, potentially controlled by gradient and/or water flow velocity (Fig. 7g). In addition to these formally described wrinkle structures, similar structures have been observed to merge with microterrace structures at sites in County Sligo, Ireland and Eastern Cape, South Africa (Fig. 7f). The terminology for terrace or barrage structures is complex, as discussed when examining the barrage macrostructure but might be resolved by separate consideration of the macro- and meso-structure. At the meso-, cm-scale, speleothem deposits have been termed 'microgours' (Hammer et al., 2010),
corresponding to 'microterrace' for travertine s.s (Bargar, 1978; Fouke et al., 2000) and for travertine *s.l.,* 'minidam' (Pentecost, 2005). Microterraces and microridges are terms applied predominantly to travertine *s.l.,* commonly travertine *s.s.* and speleothems (Pentecost, 2005).  Hammer et al. (2010) recognise that on travertine *s.l.* deposits, where the gradient is so steep that backing pools do not form, a gradation into 'microridges' normal to flow may form. These microridges are considered here to be analogous with the wrinkles described by Edwards et al. (2017). It is recommended that the terms
microterraces and microridges be applied to rock coast microbialite nomenclature, in line with other terrestrial carbonate literature and the term 'wrinkles' be abandoned. However, before these terms are applied to rock coast microbialites, care must be taken to establish biogenicity given the occurrence of morphologically similar structures in typically non-biogenic deposits. Similarly, on sub-vertical surfaces, a distinct 'pedestal'-like topography is apparent, where small groundwater channels flow over the tufa surface, with growth of island-like pedestals in between channels (Fig. 7h). Although this is an uncommon



structure it has been observed across a very wide geographic range with locations at the Eastern Cape, South Africa, Western

Australia and Ireland.

### 4.3.3 Internal organisation: stromatolites, thrombolites and leiolites

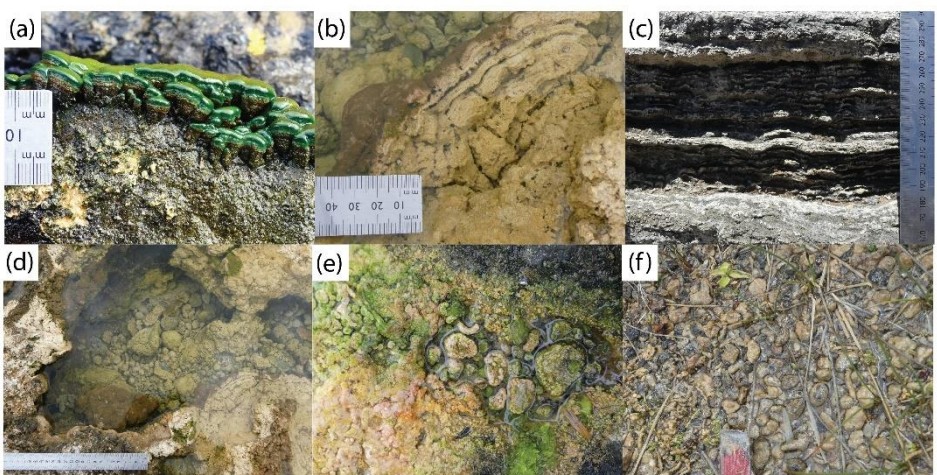

**Figure 8. Rock coast microbialite internal organisation (mesostructure): (a) stromatolitic columns coalescing to form colloform mat**
**topography (Bundoran, Ireland); (b) stromatolitic (laminar) internal organisation of a colloform stromatolite (Lauries Bay, South Africa); (c) stromatolite (laminar) barrage (Quarry Bay, Western Australia); (d) barrage pool associated oncoids; (e) fluvial crust with small oncoids and proto-oncoids (Sligo Bay, Ireland); (f) vegetation-associated oncoids (Islay, Scotland).**

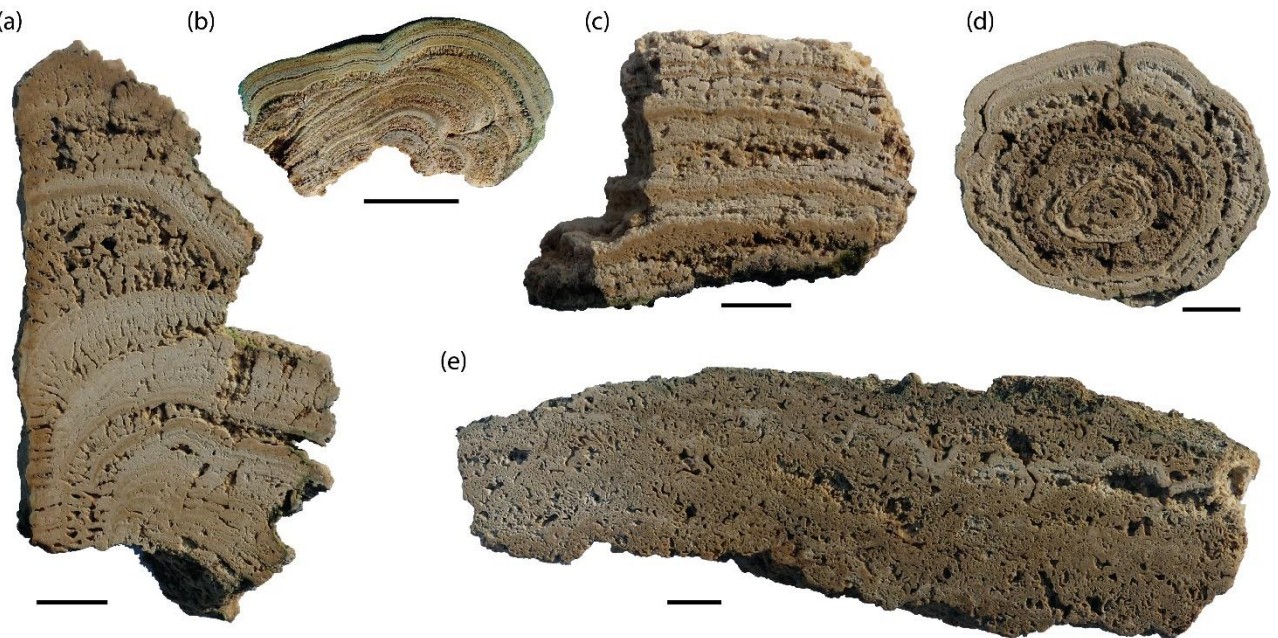



**Figure 9. Example sections of internal organisation (scale bar: 1cm): a: composite microbialite (Lauries Bay, South Africa); b: top**
**of columnar stromatolite (from field photograph a (Bundoran, Ireland); c: barrage pool-associated oncoid (from field photograph**
**D) (Lauries Bay, South Africa); d: stromatolite (laminar) (Aughris Head; Ireland); e: thrombolite with macro-laminae partially**
**preserved (Sligo Bay, Ireland).**

The dominant internal organisation of rock coast microbialites is laminated and, as such they are stromatolites (Fig. 8a-c, Fig.
9a-c). The lamination of South African stromatolites has been examined by Smith et al. (2005, 2011), describing lamined
(LLH-C ((Logan et al., 1964)) and columnar (SH-C) mesostructure, and later by Edwards et al. (2017) and Smith et al. (2018)
recognising columnar mesostructure.

Smith et al. (2011) described the rimstone mesostructure and the rimstone morphology which was subsequently recognised by
Edwards et al. (2017) as a bioturbated irregular morphology with a matted fabric that forms at barrage rims. Edwards et al.
(2017) explicitly state that lamination is not present. Rimstone tufa was also described from the barrage rims of Western
Australian sites (Forbes et al., 2010). The rimstone mesostructure is seen as intrinsically linked to the barrage macrostructure;
however, these two structures are not mutually exclusive. The well-laminated stromatolitic internal organisation within the
barrage macrostructure at Quarry Bay, Western Australia, for example, indicates that rimstone mesostructures are not
exclusively thrombolitic.


This thrombolitic internal organisation has multiple geneses related to the trapping and binding of peloids (Castro-Contreras
et al., 2014); the morphology of the microbialite builders (Kennard and James, 1986), and the disruption, destruction, and
alteration of stromatolitic internal organisation by biota: bioturbation (Walter and Heys, 1985). The latter has been suggested
as a mechanism for the formation of thrombolites in South African rock coast microbialites, predominantly by metazoans and
potentially by foraminifera and coccoids (Dodd et al., 2021; Weston et al., 2018). This may also be involved in the genesis of
leiolitic (*sensu* Braga et al., (1995)), or structureless, mesofabrics (Dodd et al., 2021). While stromatolitic, thrombolitic and
leiolitic internal organisation has been observed in the rock coast setting, future work is required to describe these structures,
and further subsets (e.g., dendrolites), before they can be confidently attributed to process.

### 4.3.4 Internal organisation: oncoids and proto-oncoids

Oncoids have not been comprehensively described from rock coast settings, however, Smith et al. (2020) describes clasts
entirely- (oncoid) and partially-encrusted (proto-oncoid) from rock pools on shore-platforms at Cape Morgan, South Africa.
These are described by Cooper et al. (2022) as a rock coast microbialite facies. These brief accounts describe barrage pool
associated oncoids; oncoids forming in the barrage pool basin of variable degrees of encapsulating lamination around a cortex
(spherical to irregular debris) (Fig. 8d, Fig. 9d). In addition, oncoids form in other supratidal groundwater settings, with
variable morphologies and characteristic environments. Fluvial crust 'rim' oncoids are oncoids and proto-oncoids with tonsure-
like rims that form around debris on fluvial crusts on bare bedrock (Fig. 8e). Proximally to the groundwater spring efflux,





vegetation associated oncoids form in paludal conditions, commonly associated with vegetation (e.g., *Phragmites australis* in the Eastern Cape, South Africa) (Fig. 8f).

Oncoids are defined by Grey and Awramik (2020) as "*unattached, generally spherical to ovoidal, stromatolite with a cortex of encapsulating or nearly encapsulating laminae*". They are considered a form of tufa or skeletal microbialite and, as such, have distinct biofabrics (Pentecost, 2005), frequently formed by cyanobacteria, in which case they may be termed cyanoid (Riding, 1991). The term proto-oncoid has been applied to clasts with partially encapsulating laminae although this term is not widely applied (e.g., Andrews and Trewin, 2014; Rodríguez and Cózar, 1999). They form within fluvial and lacustrine settings

alongside other microbialite constituents such as microbial crusts, and are commonly considered to be indicators of agitated water, with the turbulence rotating and overturning cortexes forming encapsulating lamination. However, in situ growth of oncoids has also been described (Lencina et al., 2023). As such, oncoids are frequently described as a constituent of the terrestrial carbonate models or classification schemes such as the braided fluviatile model (Pedley, 1990) and fluvial and lacustrine crust (Pentecost and Viles, 1994). While these are considered in this morphological classifcation of rock coast

microbialites to equate to macrostructures, oncoids are frequently described as having macro- and meso-structure (e.g., Casanova, 1994; Lencina et al., 2023; Villafañe et al., 2021); hence, a flexible approach to the structural level should be considered.

### 4.3.5 Microbial mat features

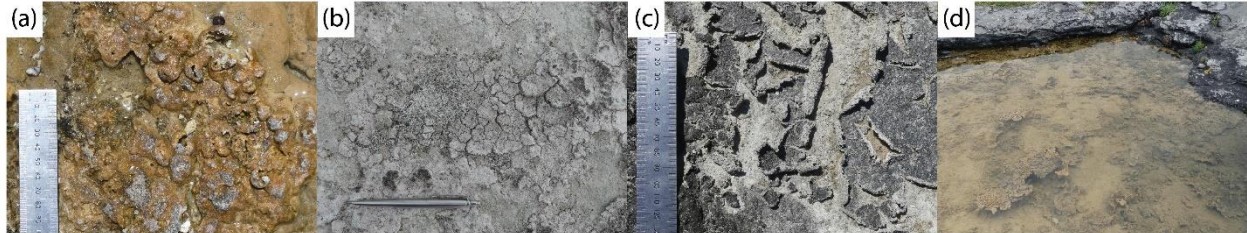

**Figure 10. Rock coast microbialite mat features: (a) Gas domes (Quarry Bay; Western Australia); (b)-(c): microbial shrinkage cracks and mat curls (Sligo Bay; Ireland); (d) microbial mat chips (Sligo Bay; Ireland).**

Microbial mats are ubiquitous in the rock coast setting and where they do not readily accrete they may develop MISS-like features, rather than stromatolites. Observations to date show that a spectrum of MISS, microbial mat features and stromatolite mesostructures are present on rock coasts and they exhibit varying degrees of stacking/accretion and mineral precipitation.


Few microbial mat features have been described explicitly from rock coast microbialites, however, the 'blister mat' mesostructure described by Edwards et al. (2017) as "*1-3cm wide dome-like structures* (that) *appear to have burst open at the top resulting in a surface morphology that resembles blistering*" may be considered to resemble burst gas domes (Fig. 10).



Blister mats accrete on low gradient substrates subject to frequent subaerial exposure (Dodd et al., 2021; Edwards et al., 2017)
and Edwards et al. (2017) suggest that their formation is a result of subaerial exposure of microbial mats, splitting domal
structures and releasing trapped gases produced by the mat decomposition, as described by Gerdesi and Krumbein (1994).
These structures are similar to the gas domes described by Noffke (2010) which can blister or collapse, releasing the trapped
gas and resulting in a crater-like morphology. These structures may be recolonised by microbial mats (Lakhdar et al., 2020)
potentially resulting in laminoid and irregular fenestrae in section as suggested by Dodd et al. (2021).


Smith et al. (2020) describe wave rip-up clasts of stromatolite within rock pools and areas stabilised by vegetation on a shore
platform setting on the South African coast, that form due to increased wave energy and/or water level during storm conditions.
These deposits may accumulate and form an allochthonous tufa microbialite breccia (Cooper et al., 2022; Smith et al., 2020).
These wave-rip-up clasts are common on shore platform surfaces where deposits of laminar stromatolites or microbial mats
are most common (Cooper et al., 2022; Smith et al., 2020) (Fig. 10d). The descriptions of these features are similar to MISS
due to comparable environmental interactions; however, due to mineral precipitation they cannot be classified as such. These
should instead be regarded as microbial mat features. Wave-rip-up clasts are comparable to MISS microbial mat chip, referring
to chip and flake-shaped centimetre-sized fragments dettached by mechanical erosion of a parent microbial mat through water
agitation, predominantly by waves or bottom currents during tides or storm events as well as potentially wind action (Eriksson
et al., 2007b; Noffke et al., 2001).

While microbial mat chips have been described from rock coast microbialite systems, microbial shrinkage cracks, also called
synaeresis cracks or dessication cracks, despite their ubiquity, have not. Microbial shrinkage cracks are a *"surface of thin
microbial mats marked with isolated lenticular, sinuously curved and even subcircular cracks, spindle shaped and tri-radiate*
*shrinkage cracks"* (Eriksson et al., 2007b) formed due to subaerial dessication in the upper intertidal to lower supratidal zones
of tidal flats (Eriksson et al., 2007b). These cracks may have curled margins, that may mature to flipped-over edges and
eventually may become a rolled-up mat fragment (Fig. 10b-c). Although these microbial shrinkage cracks and mat curls have
not yet been formally described in detail, their presence warrants inclusion in this scheme.

### 4.4 Microscale: microstructure and microfabric

Microstructure was initially described as 'the fine-scale structure of the stromatolite lamination, in particular the distinctness,
continuity, thickness and composition of the laminae' (Preiss, 1974); however, this definition has been subsequently refined
with some aspects of laminae architecture having been redefined as mesostructure by Grey and Awramik (2020).
Microstructure is considered here to refer to features visible under the microscope and includes texture, fabric, and
microfossils/microorganisms (Grey and Awramik, 2020). It has been proposed as a classifier for microbialites; however, this
has not been universally accepted due to doubt over the consistency and concurrency of microstructure preservation through



diagensis (Grey and Awramik, 2020; Semikhatov and Raaben, 2000). Therefore, Grey and Awramik (2020) suggest that microstructure and other associated features (e.g., petrography, interspace filling and diagenetic modification) should be described thoroughly, providing definitions for various commonly applied microstructural terms.

Rock coast microbialite microstructure has been described from South African localities, first by Smith and Uken (2003) and Smith et al. (2005) and most comprehensively recently by Edwards (2019). The microstructure of West Australian and Irish localities was described by Forbes et al. (2010) and by Cooper et al. (2013), respectively. The study of rock coast microbialite microstructure has utilised teminology and classification from terrestrial carbonate research, commonly tufa (e.g., thin section analysis by Edwards (2019) applies a tufa classification by Ford and Pedley (1996), revised by Jones and Renaut (2010)).
Microstructural terminology is considered relatively consistent, with terminology applicable to a wide variety of terrestrial carbonates. For future work on rock coast microbialite microstructure there should be a focus on the application of established terminology from key literature (e.g., Flügel, 2010; Ford and Pedley, 1996; Grey and Awramik, 2020; Pentecost, 2005).

## 5 Conclusions, integration of nomenclature and recommendations for future work

Rock coast microbialite research has expanded greatly in the twenty years since the first formal description by Smith and Uken
(2003) with the terminology for characterisation borrowed from both 'classical' microbialite, 'non-classical' microbialite and terrestrial carbonate nomenclature. Therefore, an integrated rock coast microbialite classification is essential in future research on rock coast microbialite systems. We have considered the existing approaches and propose a new, integrated geomorphological terminology and classification scheme, to promote future consistency. This classification addresses the presence of synonyms, duplications, omissions, and the wide variety of terms applied to the same morphological features,
predominantly at the macro- and meso-scale and it is approached through a hierarchical scale (Grey and Awramik, 2020; Shapiro, 2005). We propose the following guidelines.

The terminology and overarching classification proposed here are focused on the geomorphology of macrostructures and mesostructures. The review of the literature, however, reveals that drivers and relationships between them are less well
understood (e.g., testing and resolving the relationships between mat topography and internal organisation). In addition to this, further investigation into the classification of global rock coast microbialite fabric and geochemistry is suggested in order to better understand their abiotic and biotic drivers.

Establishing biogenicity of rock coast microbialites and associated carbonate deposits (e.g., beachrock and rhizoliths) is also
crucial, considering the genetic definition of microbialites. While Edwards et al. (2017) acknowledge that degassing of carbon dioxide inorganically can contribute to microbialite facies, alongside microbial acivity, the respective contribution of both



carbonate-forming mechanisms is not well understood with respect to different macro- and meso-structures. Greater understanding of this may allow for better integration with both terrestrial carbonate and 'classical' microbialites.

The global distribution and variability in rock coast microbialites also requires further research, with consideration of drivers such as groundwater seep chemistry, geology/topography and microbialite builders (Rishworth et al., 2020) in order to better understand their morphogenesis. Since some of the more intricate drivers of microbialite genesis, especially regarding rock coast microbialites, are currently only partly understood, this classification scheme is out of necessity based on morphology rather than morphogenesis. While this is appropriate and relevant for the interpretation of rock coast microbialites in the fossil 760 record where the specific drivers cannot be easily observed, better understanding of the current drivers in their formation would make comparisons to deep time microbialites more meaningful.

The strong focus on British and north west European studies on tufa and travertine, especially during the establishment of classification schemes (e.g., Pentecost, 1993) diminished their global value (Pentecost and Viles, 1994). This was also 765 recognised by Jones and Renaut (2010), who state that the tufa classification of Ford and Pedley (1996) accounts for tufa deposition predominantly in a cool, temperate climate and does not sufficiently consider other (e.g. tropical) climates (Carthew et al., 2003; Shiraishi et al., 2022). A global view of rock coast microbialite classification relies upon detailed description, study and comparison of sites around the globe. As new rock coast microbialite sites are inevitably discovered and described, consistency in their description and the terminology applied is vital for advances in our understanding of these deposits.


*Author contributions*: TG proposed, developed, and wrote the manuscript draft, with contributions from JAGC, AMS, GMR and MF.

*Competing interests*: The authors declare that they have no conflict of interest.


*Acknowledgements*: The authors would like to thank the Extant Peritidal Stromatolite Network (EPStromNet) team and the Supratidal Spring-Fed Living Microbialite Ecosystems (SSLiME) team for in-field discussion and support; notably, Professor Graham J. C. Underwood, Carla Dodd and Tristin O'Connell as well as Dr Joerg Arnscheidt.

*Financial support:* This review was made possible through and the Department for the Economy (DfE) sponsorship for postgraduate studentship; with the support of the EPStromNet- Extant Peritidal Stromatolite Network NERC grant award (NERC Reference NE/V00834X/1) and the Quaternary Research Association (QRA) New Research Workers Grant.





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
