# Peer review of "Reviews and syntheses: Tufa microbialites on rocky coasts towards an integrated terminology"

_EGUsphere, 2024_

## Author Response (AR1)

**Authors response.**

Key issues raised:

- **Highlight main objectives**: raised by RC2 and AE, the main objectives have been more explicitly stated. This includes greater clarity on the aims and structure of the manuscript (see lines 53-65 of track comparison document).

1) **Improve overall organisation**: raised by RC1, RC2 and AE, the manuscript structure and organisation has been revised to remove extraneous text and to improve its organizational flow.  A paragraph describing the structure has also been added to aid the reader. This includes some restructuring and renaming of parts (see 2 and sub-divisions and 3 and sub-divisions).

2) **Revise classifications**: raised by RC2 and AE, the classifications have had some information on thickness, texture, sedimentary structures added (e.g., see lines 561- 566 of track comparison document), however, the title of this work does state that it is 'towards' an integrated terminology and this information is scant for rock coast microbialites currently. This work aims to form the basis of such descriptions that will offer future refinement.

3) **Update references**: raised by RC1, RC2 and AE, references have been checked, with some references fixed to better refer to sources (e.g., by chapter) and new updated references added.

4) **Be more accurate with expressions**: raised by RC2 and AE, expressions and concepts have all been carefully examined to ensure the correct terminology has been applied.

---

## Author Response (AR2)

**Authors response.**

Key issues raised:

1) **Overwhelming information**: extra information that is not directly relevant to the subject has been removed (e.g., see lines 137-140 of track comparison document) or reduced (e.g., see lines 168-170 of track comparison document).

2) **Objective to be stated at end of introduction**: I was slightly confused by this statement as at the end of the introduction we state the three aims of the study followed by a couple of sentences on how these are achieved (see line 55 of track comparison document).

3) **Avoid 'non-classical'**: I absolutely agree, this a relict of early work on the manuscript. All statements have been removed.

4) **Description of what a rocky coast is**: This is already partially achieved by the Cooper et al., 2022 quote (see line 67 of track comparison document) but further information has been provided (see line 69-71 of track comparison document). This is partially absent due to a lack of published information/data relating the environmental setting to the microbialite deposits and secondly due to a focus on nomenclature not an overarching review (see Rishworth et al., 2022 for this).

5) **Issues in defining microbial mats and MISS confusion:** it is evident that there was confusion in the manuscript surrounding microbial mat features and MISS. This is partially due to a lack of published work on these in a rock coast environment and partially due to the authors own lack of understanding. I have tried to clarify this and explain the difference between mat topographies and microbial mat features (see line 715 of track comparison document) and explain the lack of current understanding.

   In addition, I have attempted to rectify any confusion between terms e.g., gas domes vs blister mat, changing photo in figure 10b.

6) **Scale and figure modifications:** scale has been added to all figures, and clarifications made (e.g., figure 3 modern examples?)

7) **Microstructures:** this section has been omitted for clarity.

8) **Summary citations:** citations have been removed.

I have attempted to address all the issues raised. I have not added a figure displaying the types of rock coast microbialite (microbial mat, stromatolite, thrombolite) as while I think this could be beneficial, I don't think it is essential (if it is a requirement to acceptance this can be made).